
# Decadal evolution of ship emissions in China from 2004 to 2013 by using an integrated AIS-based approach and projection to 2040

**Cheng Li[1], Jens Borken-Kleefeld[2], Junyu Zheng[1], Zibing Yuan[3], Jiamin Ou[4], Yue Li[5], Yanlong Wang[3], Yuanqian Xu[3]**

[1]Institute for Environmental and Climate Research, Jinan University, Guangzhou 511442, China
[2]The International Institute for Applied Systems Analysis, Air Quality and Greenhouse Gases Program, 2361 Laxenburg, Austria
[3]School of Environment and Energy, South China University of Technology, Guangzhou 510006, China
[4]School of International Development, University of East Anglia, Norwich, NR4 7TJ, United Kingdom
[5]Transport Planning and Research Institute, Ministry of Transport No.2 Building, 6A Shuguangxili, Chaoyang District, Beijing 100028, China

*Correspondence to:* J.Y. Zheng (zhengjunyu_work@hotmail.com)

**Abstract.** Ship emissions contribute significantly to air pollution and pose health risks to residents of coastal areas in China, but the current accounting remains incomplete and coarse due to data availability and inaccuracy in estimation method. In this study, an Automatic Identification System (AIS)-based integrated approach was developed to address this problem. This approach utilized detailed information from AIS and cargo turnover and the number of vessels calling information, thereby capable of quantifying sectoral contributions by fuel types and emissions from ports, rivers, coastal and over-the-horizon ship traffic. Based upon the established methodology, ship emissions in China from 2004 to 2013 were estimated, and those to 2040 in every five year interval under different control scenarios were projected. Results showed that for the area within 200 nautical miles (Nm) of the Chinese coast, $SO_2$, $NO_X$, CO, $PM_{10}$, $PM_{2.5}$, and hydrocarbon (HC) emissions in 2013 were 1,010, 1,443, 118, 107, 87 and 67 kt/yr, respectively, which doubled over these ten years. Ship source contributed ~10% to the total $SO_2$ and NOx emissions in the coastal provinces of China. Emissions from the proposed Domestic Emission Control Areas (DECAs) within 12 Nm constituted approximately 40% of the all ship emissions along the Chinese coast, and this percentage would double when the scope is extended to 100 Nm. Ship emissions in ports accounted for about one quarter of the total emissions within 200 Nm, within which nearly 80% of the emissions were concentrated in the top ten busiest ports of China. $SO_2$ emissions could be reduced by 80% in 2020 under 0.5% global sulfur cap policy. In comparison, a similar reduction of NOx emissions would require significant technological change and would likely take several decades. This study provides solid scientific support for ship emissions control policy-making in China. It is suggested to investigate and monitor the emissions from the shipping sector in more detail in the future.





## 1. Introduction

Although more than 30% reduction in ambient $PM_{2.5}$ levels have been achieved during the past several years in major city clusters in China due to stringent control measures, the ambient $PM_{2.5}$ levels are still far higher than the WHO Air Quality Guidelines of 10 $g/m^3$ annual average. Strengthened

reduction efforts are needed to reduce the adverse impact of ambient $PM_{2.5}$ on public health. In comparison with tightened controls on power plants, industry and road vehicle sectors, controls on ship emissions, one of the most significant contributors to ambient $PM_{2.5}$ pollution along the river and in coastal areas (Liu *et al.*, 2016), are still lax in China. Ship emissions control have been put on the agenda of $PM_{2.5}$ reduction in the coming years (Ye, 2014; Yang *et al.*, 2015; Fu *et al.*, 2016).

However, estimation of ship emissions in China remain incomplete and largely inaccurate. Locally, ship emission inventories are generally compiled in limited provinces and ports (Fu *et al.*, 2012; Bao *et al.*, 2014; Song, 2014; Tan *et al.*, 2014; Yang *et al.*, 2015), while in global inventories, ship emissions from China are of coarse temporal (monthly) and spatial (1°×1°) resolutions (Endresen *et al.*, 2003; Corbett *et al.*, 2007; Paxian *et al.*, 2010). A recent study develops ship emission inventory

in Asia with spatial resolution of 3 km×3 km (Liu *et al.*, 2016), however, characteristics of coastal and ship traffic emissions, sector-based contributions in Chinese ports and their temporal characteristics remain unknown. Furthermore, estimates from domestic (Fan *et al.*, 2016; Li *et al.*, 2016) and international studies (Endresen *et al.*, 2003, 2007; Corbett *et al.*, 2003, 2007) are associated with large uncertainties due to inconsistency in estimation approaches and data sources, thus hampering the

formulation of an effective ship emissions control strategy. Therefore, detailed and reliable ship emission inventories are needed in estimating potentials of ship emission reduction and the formulation of air pollution and public health improvement strategies.

Automatic Identification System (AIS) data, an automatic vessel position reporting system, has been widely recognized as a reliable data source that can significantly reduce the uncertainty in ship

activities and their geographic distribution (Wang *et al.*, 2008; Dalsoren *et al.*, 2009; Bandemehr *et al.*, 2015). The accuracy of ship emissions estimates based on cargo volumes (Schrooten *et al.*, 2009) and vessel arrived numbers (Yang *et al.*, 2007; Yau *et al.*, 2012; Li *et al.*, 2016) can be improved by using AIS data. Recent studies use AIS data to estimate emissions from all ships at a given time or each single trip in an entire year in Asia and Europe (Liu *et al.*, 2016; Jalkanen *et al.*, 2016). However, the

entire AIS database is not freely available to the public, especially in East Asia, and the data before 2012 is not suitable for use due to significant absence of data from a limited number of satellites and shore-based radars (He *et al.*, 2013). Therefore, establishing an integrated ship emission estimation and validation approach capable of handling incomplete AIS dataset is essential to enhance the usage



of AIS data. This integrated approach improves temporal resolution and accuracy of ship emission estimates by cross-validation between port-based and cargo-based methods, thereby providing more detailed estimations on sector-based contribution of fuel consumption and emission of air pollutants.

This integrated approach is also capable of estimating historical decadal evolution of ship emissions,

A few studies indicate that ship emissions in China has more than doubled over the last decade (Liu *et al.*, 2016), and their evolutions in the future decades are certainly of great interest to atmospheric science community and policy-makers, as development of ship emissions control policy, e.g. Chinese Domestic Emission Control Areas (DECAs) (Ministry of Transport of China, 2015) and 0.5% global sulfur cap (International Maritime Organization (IMO), 2016), profoundly impact both domestic

shipping sectors and international trade stakeholders, however, these decadal ship emission data are currently unavailable in inter-annual trends of $SO_2$ (Lu *et al.*, 2010) and NOx (Zhao *et al*., 2013) emissions from anthropogenic sources and the Multi-resolution Emission Inventory for China (MEIC) (He *et al.*, 2015). Rebuilding historical ship emission data will not only address data gap in current emission inventories, but also can help forecast future port and ship emissions, and assess the

effectiveness of ship emission control measures.

In this study, we illustrated development of the integrated AIS-based ship emission estimation and validation approach by combining detailed information in cargo turnover and the number of vessels calling. Emissions from river vessels, ports, and ocean-going vessels (OGVs) up to a distance of 200 nautical miles (Nm) from the Chinese coast were calculated. Estimations during 2004-2013 were used

as a basis for projections upon different control scenarios at five-year intervals until 2040. Current legislation and the DECAs policy were factored into the scenarios. $SO_2$ and NOx emission reductions by additional emissions control policies on ships and ports were evaluated. This study demonstrates the first effort in estimating national-scale ship emissions in China with improved accuracy by combining information of port-based vessel arrived numbers and province-based cargo volume.

**2. Approach and data sources**

### 2.1. Domain and ship categorization

The study domain includes all ports in China and offshore waters within 200 Nm of the coast (17.93 to 41.82°N, 105.28 to 124.43°E). Based on the proposed DECAs and approved global ECAs approved by IMO (IMO, 2010; 2016), ship emissions within 12 and 200 Nm from the coastline of China,

excluding offshore islands, were estimated (Fig. 1). To identify the transport distance and activity time-in-modes for ship emissions estimations, six port groups were defined geographically, namely Bohai, Shandong Province, Yangtze River Delta (YRD), Western Taiwan Strait, Pearl River Delta (PRD) and



Beibu Gulf. The reason for choosing the 200 Nm offshore as a research domain is that we want to assess how the setting of ECAs influences on emission reductions from shipping sources. More information about the domain is presented in the Supporting Information (SI) Tables SI-1 and SI-2.

In this study, ships were classified by three classification schemes, as detailed in Table SI-3. Four sub-
categories were classified by ship types, i.e. cargo ship, container, tanker, and others. Three sub-categories were classified by ship flag and customs declarations to the Marine Department (MD) of China, i.e. OGVs operated under a foreign flag or engaged in international trade, coastal vessels (CVs) operated under the Chinese flag and not engaged in international trade, and river vessels (RVs) operated in the rivers and statistically independent in local MDs. Three sub-categories were classified
by operational modes, i.e. at sea, maneuvering and at berth (IMO, 2014), as indicated in Table SI-4. Emissions from the main engine (ME), auxiliary engine (AE) and auxiliary boilers (AB) were considered, but ships that only traveled through the domain but did not call at a Mainland China port were not included.

### 2.2.   Approach

*2.2.1.   Estimation approaches*

Different emission estimation approaches were described for shipping emission inventories worldwide (Dalsoren *et al*., 2009), in East Asia (Liu *et al*., 2016), and on a regional scale (Li *et al*., 2016). In this study, an AIS-based integrated approach were established to identify emissions contributions and their historical trends. This approach integrated two AIS-based methods to address the problems of data
availability and completeness. One is the port-based approach, which makes use of the AIS-based ship activity time-in-mode in 2013 to fill the data gap in port-based vessel calling number. This enables a detailed characterization of ship emissions and their uncertainties. The other is the cargo-based approach which used province-specific cargo volume data categorized by cargo type and trade type. By combining this activity data with the average distances of major navigation routes between ports
obtained from AIS-based digital map, the historical emissions can be estimated. The cargo-based approach considers the effects of trade type, ship type structure, fuel quality and port function on ship emissions. Meanwhile, cross-validation between different statistical methods ensures robust and reliable inventory estimates. Detailed of two approaches were introduced in the following sections.

**1) Port-based approach**

The port-based approach calculates ship emissions based on engine activity, as shown by Eq. (1) (U.S.EPA, 2000, 2008; Ng *et al.*, 2012):

$$E_k = \sum_{i=1}^{n} VAN_i \times P_{lj} \times LF_{ljm} \times T_{ljm} \times EF_{ljk} \tag{1}$$



where *i, j, k, l, m,* and *n* represents a single voyage, engine type, pollutant, dead weight tonnage (DWT) class in ship type, activity mode, and total vessel arrived number, respectively. *E* is emission (g), *VAN* is the vessel arrived number, *P* is the average installed engine power (kW), *LF* is the average engine load factor, *T* is the average operation time in three activity modes (h), and *EF* is the emissions factor

corresponding to the engine and fuel types (g/kW·h).

In this equation, *VAN* were further divided into sub-categories according to ship type and DWT. The engine power and load factors of the ME were estimated using the Propeller Law based on the relationship between the instantaneous speed and the design speed, together with the detailed technical information of ship engine which was widely used in the estimation of ship emissions (ICF

international, 2009). Owing to the lack of information and similar propulsion ratios for AE and AB (Ng *et al.*, 2012), the propulsion ratios and load factors for different ship types and operational modes were obtained from technical reports (U.S.EPA, 2008; Starcrest Consulting Group, 2009). Due to the difference in the DWT sub-class range and distribution of ship profile in different studies, adjustments were made for major ship types such that the AE and AB engine defaults better corresponded with the

ship size and tonnage (Entec UK Limited, 2010; Ng *et al*. 2012). In addition, AE was assumed to be off when the ship speed was more than 8 knots (except for container and passenger ships), and those ships with diesel-electric engines were assumed not to use their boilers.

An AIS-based ship trajectory was used to define the cruising time and maneuvering time for OGVs and CVs, which included the position, time, status, speed and course of ship. For RVs, the cruising

time was calculated as the average transport distance divided by average ship speed, and considering the main navigation routes in different regions. The hoteling time can be calculated using publicly available data regarding ship activity in the main ports of China, such as the ship name, ship type, destination harbors, departure times and arrival times (http://www.chinaports.com/). The calculation results are listed in Table 1. More information regarding emissions estimations is presented in the SI.

**2) Cargo-based approach**

A cargo-based approach considering the fuel consumption rate and transport distance is shown in Eq. (2):

$$E_k = \sum_{l=1}^{n} Q_{lr} \times TD_r \times F_{lm} \times EF_k \times 10^{-6} \tag{2}$$

where *l, k, r,* and *m* is ship type, pollutant, activity region and fuel type, respectively. *Q* is the transport

volume (kt); *TD* is the average transport distance along the main navigation route (Nm); *F* is the fuel consumption rate (kg of fuel /kt·Nm); and *EF* is the emissions factor (g/kg of fuel).





Among these parameters, the stock of waterway cargo types in different provinces was separated into OGVs, CVs and RVs using the province-specific throughput of coastal ports and river ports, and it was then adjusted by the contributions of foreign trade in the main ports. In addition, the average transport distance was measured according to the historical AIS-based ship trajectories on a digital map (Fig. SI-1). It should be noted that for CVs and RVs, the value was calculated using waterway cargo throughput, and the AIS historical data for different port clusters were consistent with the statistics from the Chinese National Statistics Bureau. The calculation results are presented in Table 2.

### 2.2.2. Temporal and spatial allocation

Ship emissions were temporally and spatially allocated using surrogates from AIS data and other official statistics. Because ship emissions are significantly correlated with port throughput and main navigation routes, the average monthly throughput data in 2010-2013 were used to depict monthly variations in emissions from container ships and cargo ships. Diurnal profiles of emissions were developed according to AIS ship track data. To minimize the bias associated with seasonal transitions and locations, sampling was conducted one day per month, with the sampling grid cells covering different water areas.

A dot-density-weighted algorithm was applied for the spatial allocation of emissions. This algorithm used the density of data "dots" to calculate spatial surrogates by weighting emissions from different pollutant types and navigation modes. The emissions in different ports and water areas were defined based on hotelling and cruising information. According to the weights of the above spatial surrogates in every grid cell, the ship emissions were distributed in 3 km × 3 km grid cells covering the research domain.

### 2.2.3. Uncertainty

Previous studies indicated that the uncertainties in ship emissions were mainly introduced by time-in-mode, load factors and emission factors (Yang *et al*, 2007; Ng *et al*, 2013), but these uncertainties were not quantified. In this study, AIS data were used to quantitatively characterize the uncertainties associated with time-in-modes and load factors using a bootstrap simulation approach. Statistical methods and expert judgment were used to estimate uncertainties in the emission factors. The uncertainty ranges of emission factors and time-in-modes are presented in Tables SI-5 and SI-6, respectively. Using Monte Carlo simulation approach, the propagation of uncertainties in the above inputs into the estimated results were evaluated. This quantitative assessment revealed key contributors of uncertainties, which called for attention for further inventory improvement and refinement.



### 2.3. Data sources and validation

#### 2.3.1. Port distribution and ship activity

To ensure the reliability and precision of the emissions estimates, verification of the input data is important. Here, the differences in input data from different data sources were analyzed, and the

potential reasons for the variations were discussed. Specifically, data at the national-level, provincial-level and port-level from different statistical departments, e.g. National Bureau of Statistics, MD and Port Association, were compared. Other parameters in the port-based approach, including vessel call, ship type stock, engine power, load factor and activity time-in-mode, were also examined.

Table 3 lists the vessel calls based on MDs. The difference between the statistics provided by the

national MD and some local MDs might be caused by the existing differences in statistical methods and different classifications of vessel calls, e.g., regular shipment, international trade, domestic trade, and local shipment. To address these differences, port-based vessel calls were summarized based on 11 regional MDs defined by the Ministry of Transport of China (MD Report, 2015, unpublished) using the same statistical approach. The RVs data were obtained directly and solely from the national MD.

As the ship statistics by MD were not classified based on DWT, an adaptive sampling approach based on real-time AIS data was adopted by considering port sizes and wharf structures to remove errors in individual sampling periods. A summary of the stock of ship types that navigated in different regions is presented in Fig. SI-2. The detailed data sources are shown in Fig. 2.

The activity information for different ship types was collected from various sources. Information for

more than 5,000 OGVs and CVs was acquired from Lloyd's Register of Ships (LRS). Registration information regarding nearly 7,600 RVs was acquired from local MDs. Liu *et al*. (2016) reported that almost 18,000 ships navigated in the East Asian Sea, which supported the representativeness of samples used in this study. The LRS and MD registration databases provided the registration number, ship type and tonnage, major sea routes, fuel type, engine information, and other relevant information

for emissions estimates.

Specifically, approximately 700 AIS-based navigation trajectories from 2013 were collected, including 350 million AIS messages with 3 million operation hours covering OGVs, CVs and RVs and major ship types. In comparison with the AIS dataset in East Asia for 2013 (Liu *et al*., 2016), approximately 10% of the AIS messages with continuous and complete ship trajectories were collected from LRS

agents ([www.shipxy.com](www.shipxy.com)) in this study. Based on this AIS dataset, ship activity profiles were established by considering different regions, ship types and size categories, e.g. time-in-modes, load factors and spatiotemporal surrogates. More information regarding the AIS data is presented in Table SI-7.



### 2.3.2. *Cargo transport trends and fuel consumption statistics*

Based on the cargo transport statistics, there were no significant differences between different statistical departments, such as China Port Statistics Yearbook (CPSY), China Statistics Yearbook, and Statistics Communique of China on the Traffic and Transportation Industry Development (CCTD).

Because CPSY provides both national and provincial cargo transport balances and covers OGVs, CVs and RVs on the provincial scale, the cargo transport data from cargo volume balances was adopted for estimation from 2004 to 2013. The relationship between cargo types and ship types was described in SI.

The fuel consumption rates used in this study were based on the median values of the range provided

by the IMO report (IMO, 2009), and they accounted for the differences between container, general cargo, bulk carrier and tanker ships. More information regarding fuel consumption rates is presented in Fig. SI-3.

### 2.3.3. *Emission factors*

Apart from activity data, pollutant emission factors are also imperative for emission inventory

development. Emission factors were described per kWh (Li *et al.*, 2016; Fan *et al.*, 2016; Liu *et al.*, 2016) and per kilogram fuel consumption (Jin *et al.*, 2009; Bao *et al.*, 2014) in different studies. To make emission factor units from the literature consistent and to analyze their uncertainties, a fuel consumption rate of 227 g/kWh was calculated for MHO and a rate of 217 g/kWh was calculated for marine diesel oil (MDO) and gas oil (Ng *et al.*, 2012). In China, most RV engines were produced by

Chinese manufacturers, such as Zichai, Weichai, and Guangchai. Therefore, the average value of emission factors obtained via field measurements on local ships were used in this study (Zhang *et al.*, 2015).

Given that the marine ship industry is associated with international trade and technology, there are no significant differences in ship engine emissions for OGVs. To reduce the uncertainties in emissions

estimates due to emission factors, the relationships between emission factors by pollutant and ship characteristics, such as engine type, fuel type, sulfur content and emissions standards, were identified using a quantitative assessment approach. In this study, $SO_2$ emissions were calculated using both the sulfur balance approach and the sulfur transfer rate, which were dependent upon the engine type (U.S.EPA, 2006; IMO, 2016; Fan *et al.*, 2016). As indicated in Table 4, sulfur contents of fuel

consumption for OGVs, CVs and RVs were determined by considering the global average value (IMO, 2016) and the local and national statistical values (Fan *et al.*, 2016). The global background values of sulfur contents used for estimation are shown in Fig. 3. To assess the historical and future trends of NOx emissions under control policies with different NOx emission standards, NOx emission factors





under different influence factors were determined from the IMO study (IMO, 2008) and Liu *et al.* (2016), as detailed in Table SI-8. Emissions of particulate matter, hydrocarbon (HC) and CO were determined based on the engine types and fuel types (USEPA, 2006, 2009), as shown in Table SI-9. All emission factors were selected according to local emission characteristics for navigational areas,

ship types and DWT values, as listed in Table SI-10. Low-load adjustment multipliers were applied when the load factors of ME were below 20% to account for the low combustion efficiency during low main engine loading conditions (ICF International, 2009), as indicated in Table SI-11.

### 2.3.4. *Control scenarios and factors for emission projection*

Ship emissions in China are largely associated with international trade pattern and ship engine

technology development. Future ship emissions are therefore determined by multiple factors, including trade and political (e.g. DECA, emissions standard), economic (e.g. GDP, shipbuilding industry), social (e.g. sulfur content of marine heavy oil (MHO), population), and technological (e.g. engine type, after-treatment devices). In this study, fuel consumption was used to predict baseline ship emission scenarios since there are strong associations between fuel consumptions and ship emissions (See Fig.

SI-4), and emission control scenarios were used to adjust baseline emissions under different control strategies.

Changes of ship fuel consumption in every five-year interval from 2015 to 2040 in China were estimated by the output data from a predication model with high reliability (IEA, 2016). Specifically, estimation of marine fuel consumption in 2013 was used as the base-year value. Fuel consumptions

associated with inland and coastal navigation sources (oil and gas) were used to predict fuel consumptions of CVs and RVs, whereas international marine bunkers were used to predict fuel consumptions of OGVs.

Ten scenarios were designed for $SO_2$ and NOx emissions reductions based on global sulfur cap of 0.5% by 2020 as planned by the IMO. Because NOx emission reductions depend on new engine technologies

such as exhaust gas recirculation (EGR), selective catalyst reduction (SCR), and liquefied natural gas (LNG) engines, a 20-year lifetime for ship engines was assumed for the engine renewal period. Current legislation and DECAs were factored into the scenarios. Additional emissions control policies targeting $SO_2$ and NOx emissions from vessels and ports were evaluated based on emissions reductions, including a baseline, $SO_2$-DECA (SECA) and NOx-DECA (NECA), as detailed in Table 5. Future

emissions were calculated at a 5-year interval.



## 3. Results

### 3.1. Characteristics of ship emissions in 2013

#### 3.1.1. Estimation of fuel consumption in ports and sea

In 2013, there was no ship emission control measure in China. Ship emissions were therefore largely

determined by fuel consumption. We start this section by discussing fuel consumption characteristics in ports and sea which can be indicative of and verify ship emissions in 2013.

The integrated approach is used to estimate ship fuel consumption in China in 2013, as shown in Table 6. Total fuel consumption based on port-based and cargo-based approaches exhibited a good agreement within 12 and 200 Nm to the coastline (deviation < 15%). More than 85% of MDO was

consumed within 12 Nm, and almost 80% was contributed by RVs and CVs, particularly by RVs. Conversely, only 35% of MHO was consumed within 12 Nm, and OGVs dominated its consumption. Although there were differences in the MDO estimations of CVs and RVs between two approaches, they were mainly associated with small CVs that were categorized as RVs in port under the port-based approach. Here, the results of port-based approach were used for comparison. Total fuel consumption

within 200 Nm was estimated to be 17,035 kt in 2013, within which 2,730 kt of MDO was consumed in rivers and coastal waters. Fuel consumption in the overlapping area estimated by Liu et al. (2016) was almost 25% greater than that in this study because of differences in domain size and estimation approach.

Fig. 4 showed the spatial allocation maps of MDO and MHO, which were calculated by the dot density

of AIS data from RVs and CVs within 50 Nm and from OGVs within 200 Nm of the coastline, respectively. As most MDO was consumed by RVs and low-power CVs, the spatial distribution of MDO follows the coastline and rivers, especially in the YRD region. OGVs predominantly consume MHO, therefore the highest densities of MHO appear in the development areas of international trade and near the international navigation routes, such as the YRD, PRD, Bohai, and regular routes

connecting YRD and PRD.

We further examined the port activity and fuel consumption for the top ten ports in China, as detailed in Table 7. The results indicated that Shanghai and Ningbo-Zhoushan contributed to ~28% of total MHO consumption, Hong Kong, Shenzhen and Guangzhou contributed 23%, whereas 36% of total MHO consumed outside the top ten ports. MHO consumption in all ports accounted for ~30% of the

total ship fuel consumption within 12 Nm. In comparison, nearly 70% of the total MDO was consumed within 12 Nm of the top ten ports, and 42% was consumed in ports. Shanghai, Guangzhou and Suzhou were the largest MDO consumption ports in China (43% of the total MDO), as a great number of RVs



were operated in the dense waterways of the YRD and PRD. It is interesting to note that the ranks of ship fuel consumption were not the same as those of cargo throughput, container throughput and vessel arrived number. The difference was mainly caused by port conditions and target clients, which further causes huge differences in emissions from different ship types. Using cargo throughput, container

throughput or vessel arrived number to represent ship emissions would therefore generate misleading results. Our results suggest that the consideration of ship type and DWT is crucial for accurate estimation of ship emissions.

Previous studies reported a strong correlation between emissions and the distance from the coastline in the YRD region (Fan *et al*., 2016; Liu *et al*., 2016). In this study, by taking advantage of the port-

based approach, fuel consumption of ocean traffic can be determined in a designated port, and fuel consumption from different coastal port clusters can be identified. As shown in Fig. 5(a), MHO consumption from the YRD, PRD and Bohai regions accounted for more than 85% of the total consumption in China, with YRD itself of 46%. This provides solid evidence for the DECAs in these three regions proposed by the Chinese government in 2016. Fig. 5(b) shows the cumulative distribution

of MHO with the distance to the coastline, which indicated that the DECAs within 12 Nm covered about 40% of the MHO from the total ship sector within 200 Nm, and can reach 80% when the distance was extended to 100 Nm.

### 3.1.2. Compilation of ship emission inventory and uncertainty

Based on the above fuel consumption results, ship emission inventory in 2013 in China were calculated

by combining with fuel-based emission factor. Table 8 lists ship emissions within 200 Nm of the coastline in 2013. Emissions of $SO_2$, NOx, $PM_{10}$, $PM_{2.5}$, CO and HC were 1,010, 1,443, 118, 107, 87 and 67 kt/yr, respectively. Compared with the total anthropogenic emissions in MEIC (http://www.meicmodel.org/), emissions from ships accounted for about 10% of the total $SO_2$ emission and 9% of the total NOx emission from all sectors in coastal provinces. Cargo ships (general cargo

ships and dry bulk carriers), container ships and tankers (chemical tankers, gas tankers and oil tankers) were the main contributors of all pollutants, accounting for 38-42%, 37-39% and 14-17% of total pollutants emitted within 200 Nm of the coast, respectively. These results are in line with previous estimates (Liu *et al.*, 2016). The AE was responsible for 20% of $SO_2$ and NOx, similar to 26% in East Asia (Liu *et al.*, 2016) but significantly higher than the global fraction of 10% (Paxian *et al.*, 2010)

and lower than 40-60% from local ports or regions (Ng. *et al.*, 2013; Fan *et al.*, 2016; Li *et al.*, 2016). These diversified results were mainly resulted from ship navigating time in cruising mode in different research domains, and the simplifications on basic parameters of AE and AB, e.g. lower output power of cargo ships and container ships in this study than those in Liu *et al.* (2016). We also noted that RVs



contributed to 6% of NOx and 2% of SO₂ in the shipping sector, which was not reported in previous studies. The majority of ship emissions occurred during ship cursing, whereas ship emissions at berth and during maneuvering only comprised 14% and 5% of total ship emissions, respectively.

Table 9 summarizes the estimated means and the associated uncertainty ranges of pollutant-based ship emissions in 2013 using Monte Carlo methods. CO emission showed relatively large uncertainties, ranging from 109 to 143 kt in the 95% confidence interval with relative errors of -10% to 18%. In comparison, the uncertainties in SO₂ and NOx were relatively small, ranging from 991 to 1,058 kt and from 1,348 to 1,556 kt, respectively, with relative errors of -6% to 9% in the 95% confidence intervals. The high uncertainties in CO estimates were mainly caused by the differences in emission factors from different sources (USEPA, 2006; ICF International, 2009; Liu *et al.*, 2016), which varied between engine type, combustion conditions and operation modes. Overall, the uncertainties reported in this study were larger than those reported in large-scale studies (~±5%) and lower than those in small-scale studies (~±20%) (Li et al., 2016; Liu et al., 2016).

### 3.1.3. Temporal characteristics

Fig. 6 shows the monthly and diurnal variations of emissions from different ship types. Based on the temporal surrogates of container and cargo transport in the southern (south of YRD) and northern (north of YRD) port groups from 2010 to 2013, the monthly variations in container and cargo ship emissions were similar with small variations. Additionally, emissions were slightly higher in August and December and lower in February. This was mainly due to increased ship activity in the summer and winter, whereas relatively less cargo transport during the long public holiday of Spring Festival in February. These variations were generally consistent with some local studies (Ng *et al.*, 2013; Li *et al.*, 2016) but differed from Fan *et al.* (2016), which indicated that ship emissions were the highest in April and no significant differences in total emissions were observed in June, November and December. Passenger ships exhibited a bimodal monthly variation pattern, with peaks in August and December.

Ferries were the only type that exhibited significant diurnal patterns. With an hourly percentage of less than 1% at midnight and in the early morning, fuel consumption from ferries increased dramatically starting at 8 am, reached a peak at 10-11 am, then slightly declined and reached another peak at 5 pm. Fuel consumption from other ship types remained constant over the course of a day because these ships were generally used for long-distance transport and sailed at all times under the 24-hour rotation system.





### 3.1.4. *Geographic distribution and emissions intensity*

Fig. 7 shows the spatial allocation of $SO_2$ (3 km × 3 km) in the ship emission inventory in 2013, with the main ports and navigation routes highlighted. It is clear that the emission distribution is strongly consistent with the current regular navigation routes. Specifically, the lines south of YRD are more

aggregated, whereas those north of YRD were more scattered and concentrated in the ports of Dalian, Tianjin and Qingdao and the transport routes in between, as shown in Fig. SI-5. By contrast, the emissions over the PRD were concentrated on the lines to the north and in the estuary. Because the YRD region is a fast-developing international shipping center and the convergence area of the south and north waterways, the emissions were very intensive in this region.

Previous studies showed that ship emissions were commonly concentrated, and ship emissions from different geographical areas, such as traffic hubs (Fan *et al.,* 2016), ports (Ng *et al.,* 2013), coastal areas (Ng *et al.,* 2012; Goldsworthy *et al.*, 2015; Li *et al.*, 2016), and the Sea (Jalkanen *et al.*, 2009; Tournadre *et al.*, 2014; Liu *et al.*, 2016), were discussed. In this study, special analysis was conducted with regard to emissions from DECAs and typical shipping routes in China.

Table 10 presents the emission intensities of $SO_2$, NOx and $PM_{10}$ in three DECAs and along four typical shipping routes (Fig. SI-5). The results indicated that all DECAs and shipping routes contributed significantly emissions to coastal waters. Only covering 19% of the total area within 200 Nm, the DECAs and shipping routes contributed to almost 36-38% and 27-29% of total emissions, respectively. As YRD-DECA has the highest traffic concentration in East Asia, the average intensities

of $SO_2$, NOx, and $PM_{2.5}$ emissions were six times those of East China Sea (Liu et al., 2016). With three busy ports (Hong Kong, Guangzhou and Shenzhen), the intensity of PRD-DECA was approximately eight times higher than average emission intensities of the South China Sea (Liu *et al.*, 2016). A previous study indicated that emission values greater than 8 t/yr/km² were common in the busy fairways of the East China Sea (Fan *et al.*, 2016), but the values associated with traffic hubs are still

ambiguous. Table 10 presents the $SO_2$, NOx, and $PM_{2.5}$ emission intensities at four traffic hubs along shipping routes. The route between the YRD and PRD (including the Taiwan Strait) is one of the busiest sea-routes in the world, and the emission intensities were similar to those in the PRD-DECA and much greater than those of the Bohai-DECA. By contrast, the sum of emissions intensities of the other three regular routes were slightly less than the route between the YRD and PRD as they became

scattered to the Bohai, South Korea and Japan.





### 3.2. Ship emissions from 2004 to 2013

#### 3.2.1. Trends in ship activities

Fig. 8 shows the multi-year estimation of MHO consumption in main port clusters within 200 Nm offshore using a cargo-based approach from 2004 to 2013. MHO consumption in China increased from 8,040 kt in 2004 to 17,035 kt in these ten years with an annual growth rate of 9%. This change has been driven by the rapid increase in international trade (10% growth in the external trade of cargo) due to the economic boom (10% growth in GDP) during this period.

The trends in fuel consumption were in a good agreement with those in cargo and container turnover in different port groups (Fig. SI-6, SI-7), and such an agreement would persist without technological revolution on diesel engine in the future. In addition, the growth rate of top-scale port clusters (PRD and Shanghai) were relatively lower than others. Specifically, traffic in the Jiangsu and Liaoning port clusters even recorded a more than three-fold increase. Such a significant increase was largely contributed by the dramatic growth of domestic trade in China, which highlighted the urgent need for emission control on RVs and CVs. There was a slight drop in traffic in 2008 amid the general increase trend in these ten years, which was largely caused by the declined external trade in most ports in China resulted from the global economic crisis (Fig. SI-8).

#### 3.2.2. Emission trends

Fig. 9 uses $SO_2$ and $NO_x$ as examples to show the trends of ship emissions in China from 2004 to 2013. The results indicated that emissions increased first, leveled off or even decreased slightly in 2008 and 2009, and then increased rapidly afterwards. The drop in 2008 was mainly caused by decreased container turnover due to weakened international trade market during the international financial crisis. This period was followed by a rapid increase with the global economic recovery after 2009. During these ten years, seaborne trade for both cargo and container transport in China tripled, but the increase of pollutant emission were slower, e.g. 1.7 times for $SO_2$ and 2.2 times for $NO_x$. The low growth rate of $SO_2$ compared to that of $NO_x$ emissions was caused by the improvement in the sulfur content of global MHO from 3.5% to 2.7% over the past ten years (Fig. 3). In comparison, emission factors of $NO_x$ decreased slightly due to technological difficulties in further improving ship engines. The increasing trends for $SO_2$ and $NO_x$ were different from those for land-based anthropogenic sources, $SO_2$ emissions from power plants and other major sources have decreased substantially since 2005 due to the application of emissions control technologies (Lu *et al*. 2010), and $NO_x$ emissions declined continuously after 2011 (Zhao *et al.,* 2013).





### 3.3. Estimation of ship emissions during 2013-2040

#### 3.3.1. Impacts of various SO₂-DECA policies

Fig. 10(a) shows $SO_2$ emission reductions based on the global sulfur cap of 0.5%, and the results indicated that $SO_2$ emissions will be reduced by over 80% when the 0.5% sulfur cap is achieved in 2020. Emissions can be further reduced by 86%, 91% and 94% within 12, 100 and 200 Nm, respectively, by expanding DECA regions with 0.1% sulfur content in oil. These results indicated the importance of lowering the sulfur content of global marine oil.

If the 0.5% global sulfur cap fails to achieve, China could make its own effort to reduce $SO_2$ emission from ships, as shown in Fig. 10(b). The proposed DECAs policy within 12 Nm in three regions can reduce $SO_2$ emissions by over 20%. Further scenarios with different DECAs strategies were calculated from 2020 to 2040. $SO_2$ emissions can be reduced by over 50% by expanding the DECAs regions to 100 Nm of the entire Chinese coast and using 0.5% sulfur content fuel. An additional 25% reduction is expected by expanding the DECA to 200 Nm and using 0.1% sulfur content fuel. 94% of $SO_2$ emissions can be mitigated in total.

#### 3.3.2. Impacts of NOx-DECA policies

Currently, the effective approaches for limiting NOx emissions from ships depends on the development of new ship engines, such as EGR, LNG, and SCR engines. Thus, reductions in NOx emissions were associated with passive step-by-step controls if no enforcement measures were implemented for existing ships. Therefore, the assumption of a 20-year ship lifetime was used in this study. Fig. 10(c) shows future NOx emissions with or without a NOx-DECA in China. If there is no emissions control plan for 'Tier III' ship engines, the emission from engines with 'Tier II' NOx emission standards will peak in 2030, and with the elimination of 'Tier 0' ship engines, a 13% reduction in NOx emissions can be achieved by 2040. By contrast, if China implements a NOx-DECA within 200 Nm of China coastline in 2020, NOx emissions can be reduced by 80%.

## 4. Discussion

### 4.1. Implications for policy-making

This study showed a significant increase of ship emissions in China from 2004-2013, which highlighted the urgent need for effective control of ship emissions. Application of cleaner fuels and environmentally friendly ship engines are possible means to reduce ship emissions in China. This study also provided justifications for the establishment of DECAs in China.

To improve regional air quality and facilitate the structural adjustment of industry, an implementation plan for DECAs in the waters of the PRD, YRD, and Bohai regions was established in December 2015.



This was a health-based initiative that is anticipated to have positive long-term effects on those who live and work in DECAs and nearby. Shanghai as a demonstration city has observed positive impact after implementation of this policy for one year. However, many issues still may hinder successful implementation of the policy. We believe the following tasks are essential: 1) more technical

guidelines and standards regarding the exhaust emissions of ships and the use of shore power and other clean energy, e.g., supervision guidelines for DECAs, should be issued; 2) qualitative and quantitative emissions management should be improved by strengthening monitoring procedures, responsible parties and managers should be quickly spotted, and the illegal emissions of air pollutants from ships should be banned; 3) smooth communication and regional cooperation should be enhanced, e.g.,

communication with shipping and energy enterprises to increase the supply of low-sulfur fuel and offset shipping costs in cooperation with multiple environmental authorities for joint prevention and control; 4) awareness from different stakeholders should be enhanced, e.g. alleviation of community and public attention, strengthening social responsibility of governments and corporations, optimization of standardized management and service function, and investment in public health mechanisms in port

areas; and 5) future phases of emissions control policies should be formulated, e.g. enhancement of the DECAs policy from local to regional, national and continental scales based on scientific findings to formulate both short-term and long-term effective ship emission control strategies.

### 4.2. Call for more comprehensive data

We used an integrated AIS-based methodology to represent the characteristics and trends of ship

emissions in China. In this methodology, it was assumed that the empirical statistics of voyages along regular routes and in ports were correlated with the ship type and geography and that emission factors changed along with changes in oil quality and engine technology. Uncertainty in ship activity parameters and emissions factors will impact the accuracy of the emissions characteristics and trends. Discrepancies in total ship emissions existed in global-, regional- and port-scale studies. The key

reason for the emissions discrepancies was not only the uncertainty in annual activity rates and emission factors but also the quality of different data sources and variations in the assumptions underlying different methods. For example, the Ports of Los Angeles and Long Beach (*PoLa)* study assumed that the ship AE was shut down when the ship speed exceeded 8 knots (except for passenger ships; Starcrest Consulting Group, 2009; Ng *et al.*, 2013), but there were other studies assuming that

the AE worked all the time (IMO, 2015; Liu *et al.*, 2016). Additionally, BEs were used on OGVs in the IMO study but were not included in ships with diesel-electric engines in the *PoLa* study.

Moreover, the estimation of emissions factors were characterized in most studies, but some studies only considered fuel type and engine type (Fan *et al.*, 2016). Some studies also ignored missing ships



in the AIS dataset (Ng *et al.*, 2013; Li *et al.*, 2016). Due to uncoordinated control policies in different regions and the poor performance of the port environmental statistics system in China, field surveys and measurements must be conducted, more accurate local assumptions must be made and a standardized methodology for estimating ship emission inventories is needed.

Small differences in the assumptions can yield large errors in the emission estimates. To avoid this problem, we suggest maximizing the collaboration with other related entities (e.g., engine manufacturers, regulatory agencies, port authorities, vessel owners, the published literature and commercial entities) to gain a more complete and unbiased understanding, fill data gaps, and mutually validate approaches. Subsequently, plans to conduct field surveys and measurements to establish local
databases and validate these assumptions should be made. Examples include the engine operation conditions of different ship types under local navigational conditions (especially under the current national emissions reduction framework), the tendency of fuel quality and engine technology, and the integrity and accuracy of the real-time data obtained from the AIS dataset.

To establish a standardized methodology for estimation, some suggestions are proposed: 1) fill the
data gap and optimize data quality by implementing various measures, e.g. data integrity and volume checks, data longevity assessment, separation of real data from assumptions/defaults, separation of activity-based values from those based on factors or equipment, provision of valid data ranges for factors and equipment, improvement in geospatial data collection, and establishment of quality assurance and quality control (QAQC) measures; 2) standardize data collection procedures, verify the
existing results, conduct third party reviews of findings, and update existing inventories with these findings; 3) develop a regulatory framework, e.g. cross-comparison datasets, limiting interpretation errors, evaluating data quality, and performing data logging and emissions testing; and 4) conduct vessel boarding programs to collect actual vessel and operational parameters, e.g. equipment duty cycle, engine operation, fuel use and fuel switching data, main, auxiliary and boiler loads according to
mode, and operational parameters according to mode.

A robust emissions inventory is essential for planning and tracking as environment challenges broaden, thus, further refinement of ship emission inventories should be conducted to ensure regulatory emissions inventories are accurate and to track the progress of emissions reductions strategies. In addition, because coastal areas in China are densely populated, more assessment studies should be
conducted based on reliable emission inventories to develop sustainable, cost-effective, environmental and human health solutions, e.g. health risk assessments, air quality assessments, and cost-benefit evaluations of control policies.





## 5. Summary and Conclusions

We demonstrated good agreement in ship emissions estimation by AIS-based integrated approach based on different data sources, and these results provided solid evidence for better understanding national-, regional- and local-scale ship emissions in China. The results indicated that ship emissions

within 200 Nm of the Chinese coast were 1,010, 1,443, 118, 107, 87 and 67 kt/yr for $SO_2$, NOx, $PM_{10}$, $PM_{2.5}$, CO and HC in 2013, respectively. Ship emissions constituted approximately 10 % of the total NOx and $SO_2$ emissions in coastal cities. Approximately 40% of the pollutants from ships were emitted within 12 Nm of the coast, and would be doubled within a distance of 100 Nm. Therefore, the expansion of the DECAs could greatly improve the control effect. YRD, PRD and Bohai Regions

contributed 46, 27 and 15% of the total MHO emissions, respectively. Additionally, about 65% of ship emissions in all ports came from the top ten ports, which also contributed to 24% of the total emissions within 200 Nm. In addition to the proposed DECAs, more attention should be paid on the emissions along regular navigational lines near coastlines, especially the Taiwan Strait and South-North routes. Furthermore, ship emissions have doubled over the past ten years, and $SO_2$-DECA and NOx-DECA

control policies can potentially achieve >80% emission reductions in the future. For NOx, similar reductions could be achieved via strict engine emissions controls, low-sulfur fuel oil and a switch to propulsion with natural gas. However, such policies would not provide substantial benefits until 2040 because decades are needed to implement fleet-wide changes. Potential reduction efforts are of considerable regional importance because ship emissions along the Chinese coast account for almost

half of the total ship emissions in East Asia.

*Acknowledgements*

This work was supported by the Public Environmental Service Project of the Ministry of Environmental Protection of People's Republic of China (201409012), National Distinguished Young Scholar Science Fund of the National Natural Science Foundation of China (41325020), and the

Chinese National Member Organization at the International Institute for Systems Analysis, Laxenburg, Austria.



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





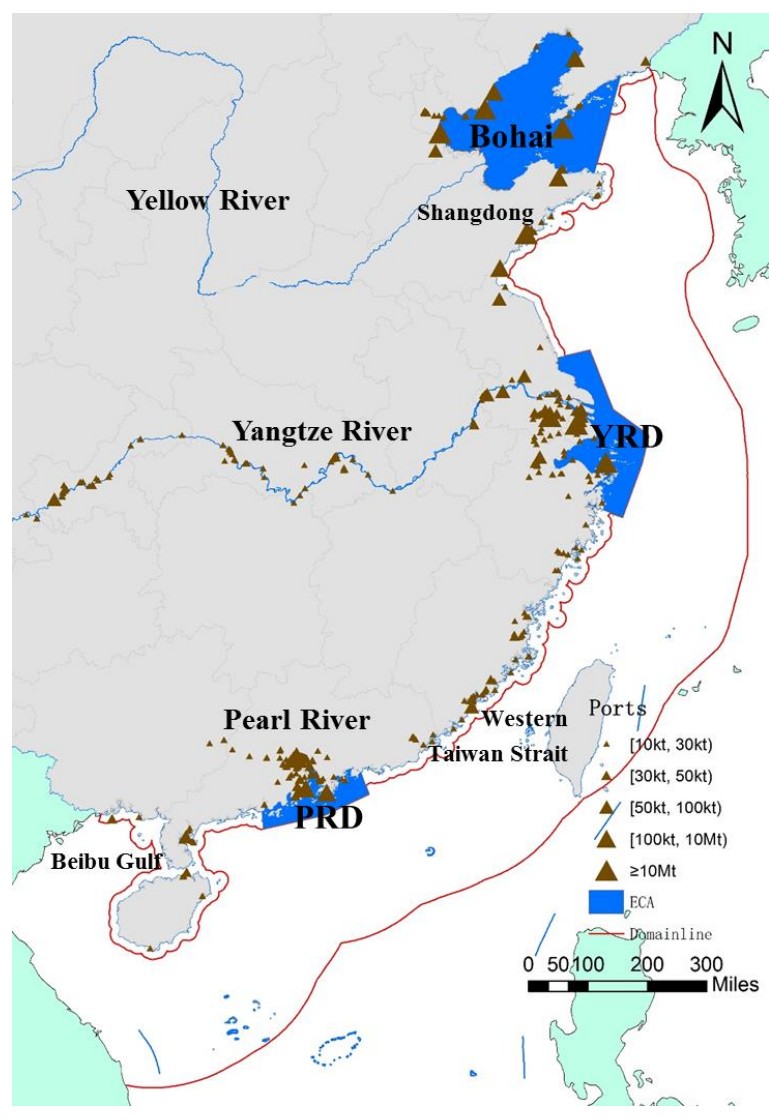

**Fig. 1 The location of the research domain, port groups and DECAs in this study**





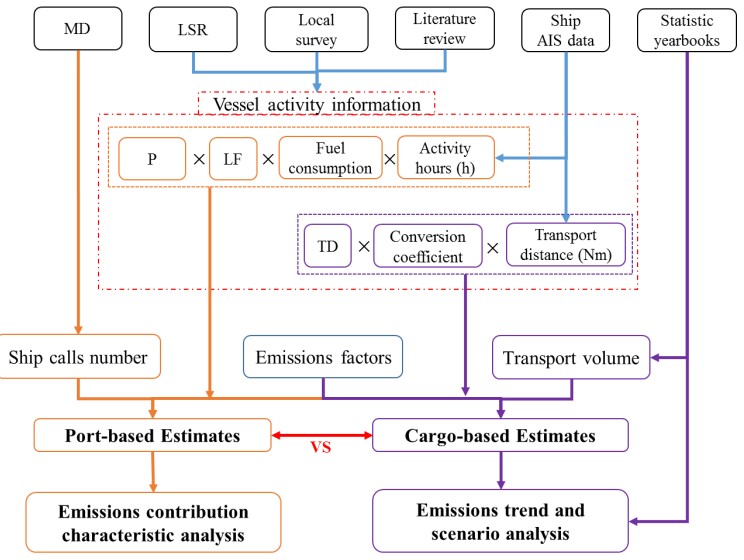

**Fig. 2 Data sources and flowchart used for emissions estimates in this study**

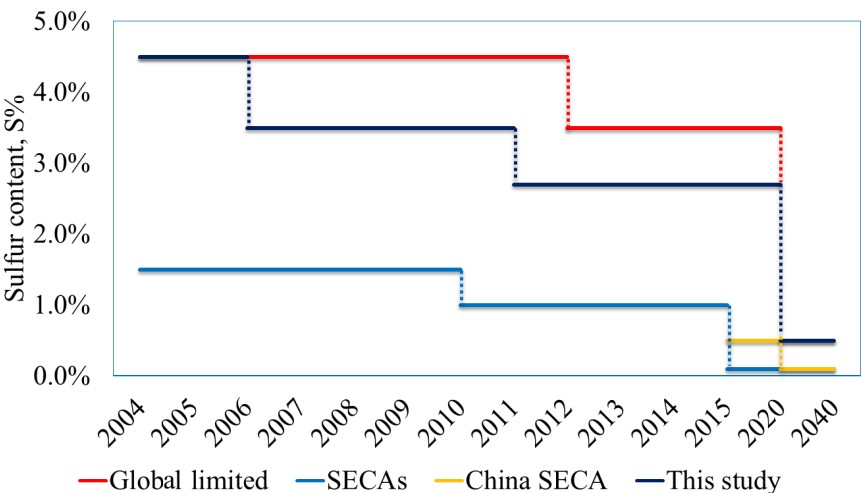

**Fig. 3 The sulfur content of MHO in this study**





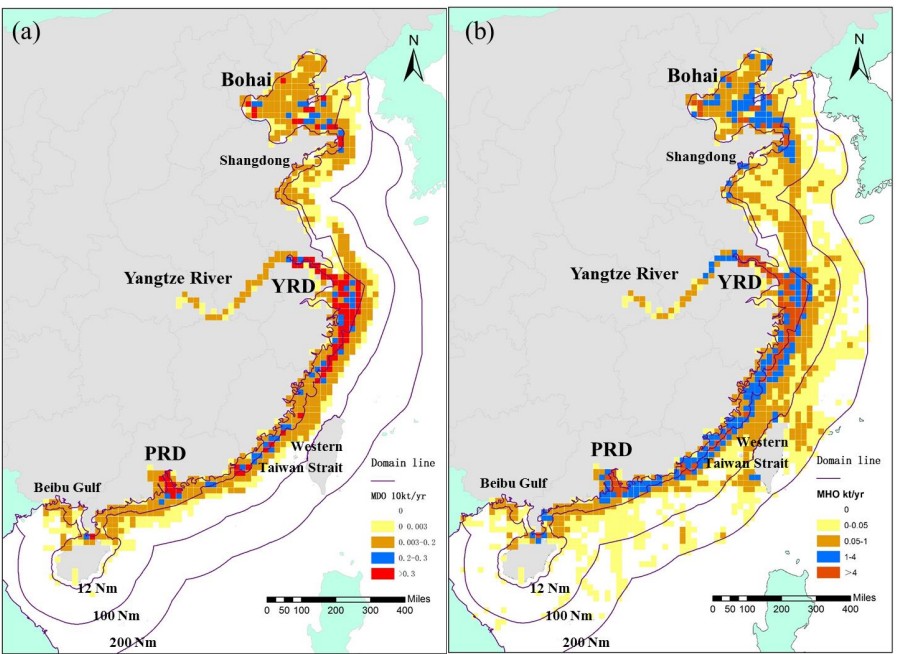

**Fig. 4 Spatial distribution of marine diesel oil MDO (a) and MHO (b) consumption by ships (27 km × 27 km)**

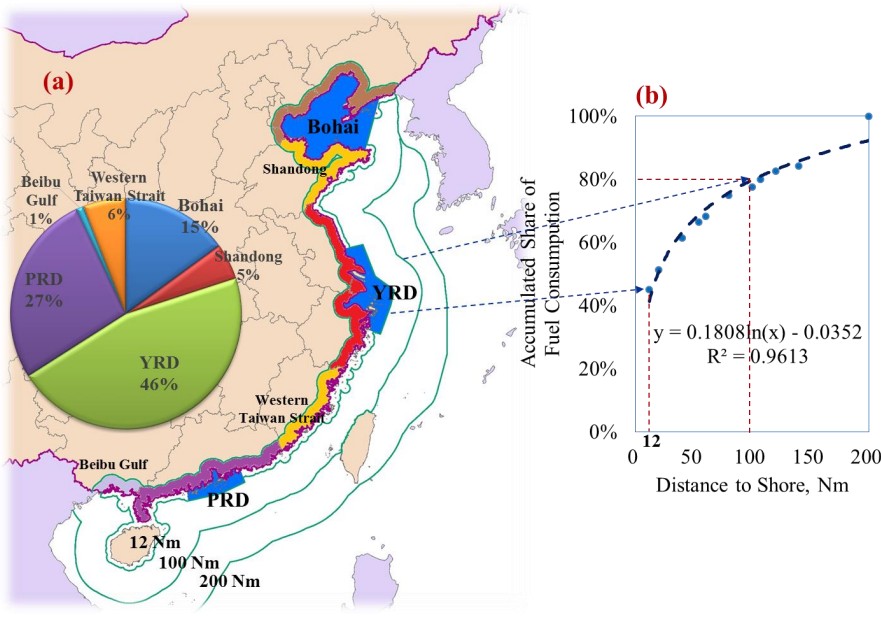

**Fig. 5 (a) Fuel consumption contributions of different port groups within 200 Nm and**
5              **(b) the cumulative distribution of fuel consumption within 200 Nm**





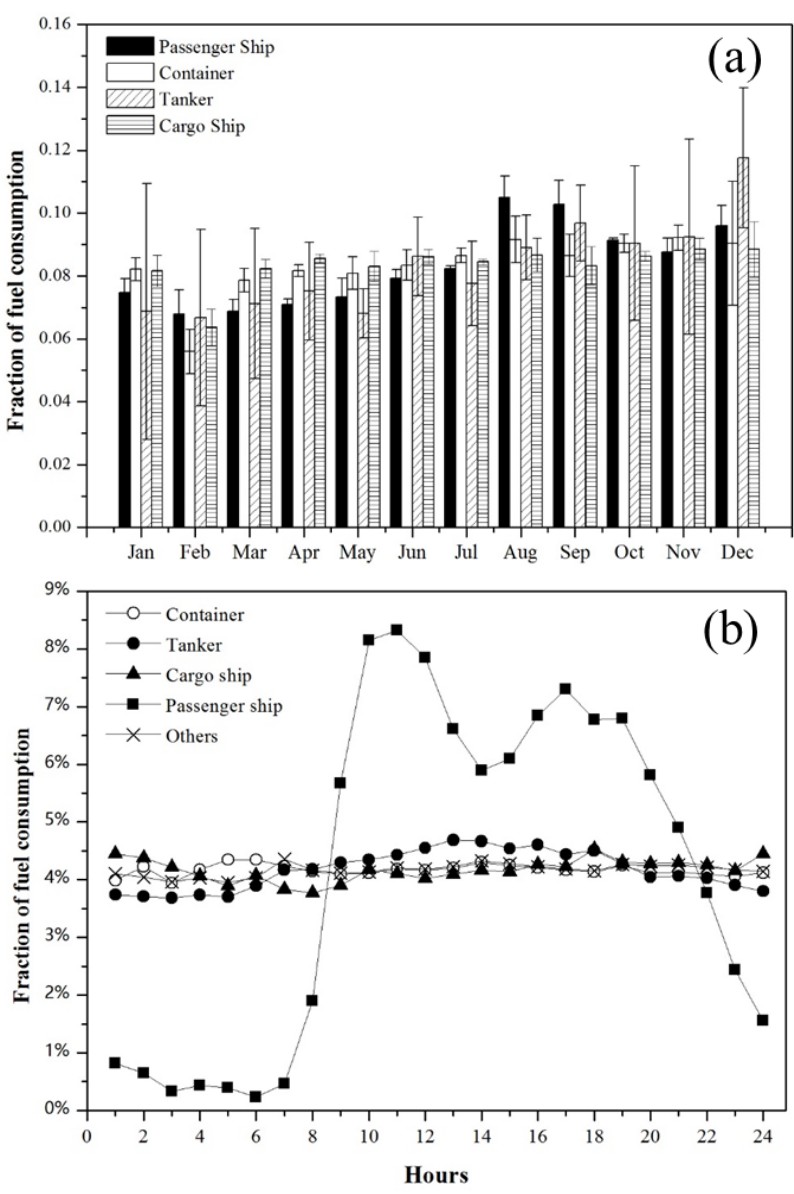

**Fig. 6 Monthly (a) and diurnal variations (b) in annual fuel consumption by vessel type**





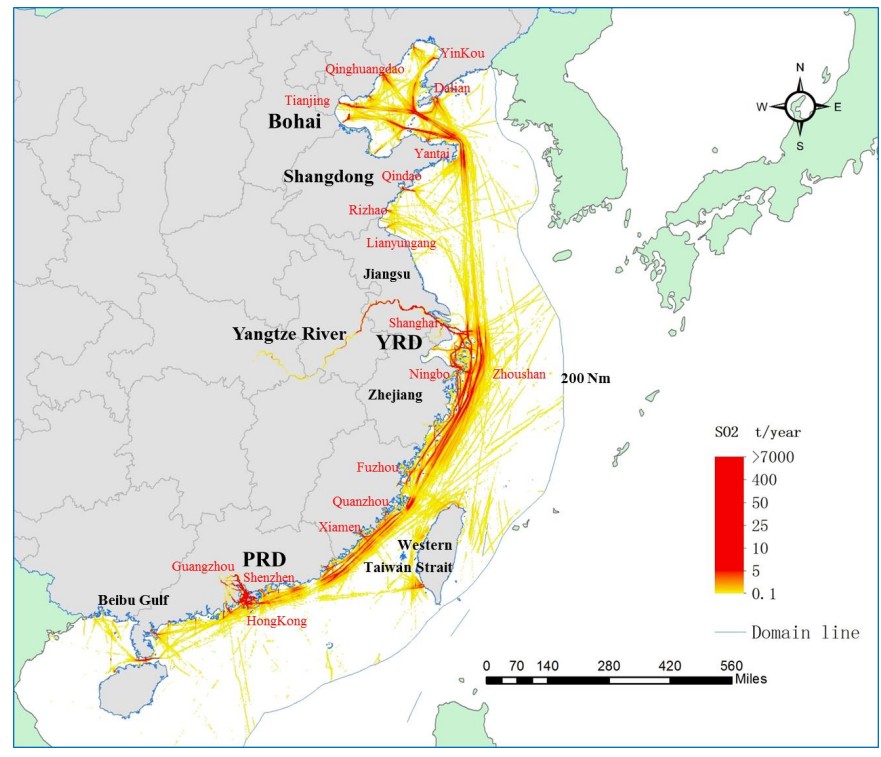

**Fig. 7 Spatial distribution of SO₂ ship emission in China (3 km × 3 km)**





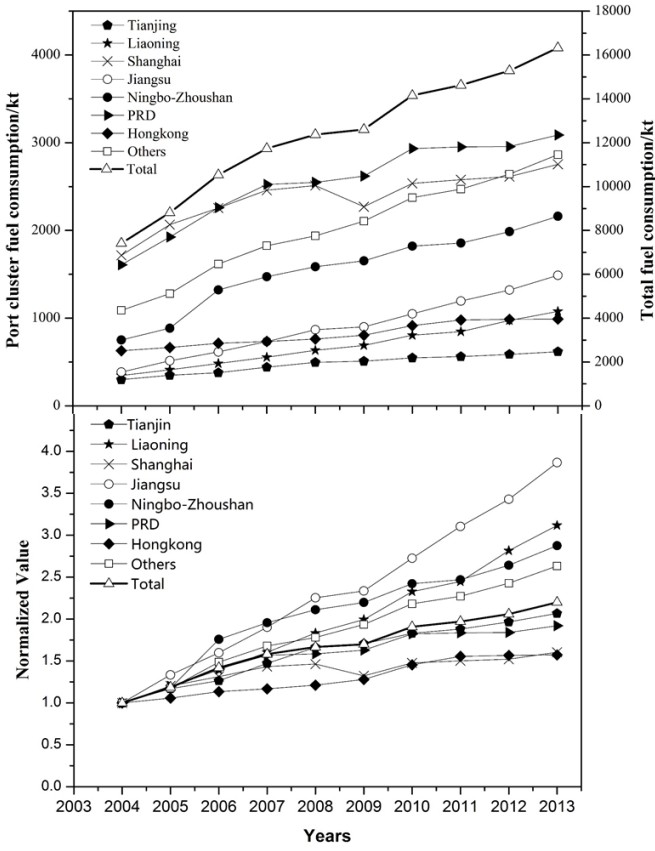

**Fig. 8 Trends in MHO consumption in port clusters in China from 2004 to 2013: (a) fuel consumption and (b) normalized fuel consumption**

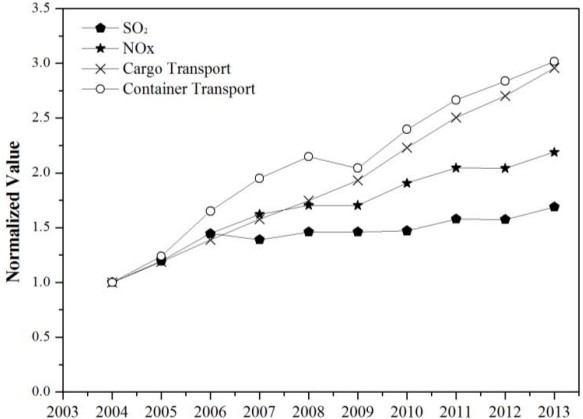

5    **Fig. 9 SO₂ and NOx emissions and their aggregated emissions from cargo and container transport from 2004 to 2013**



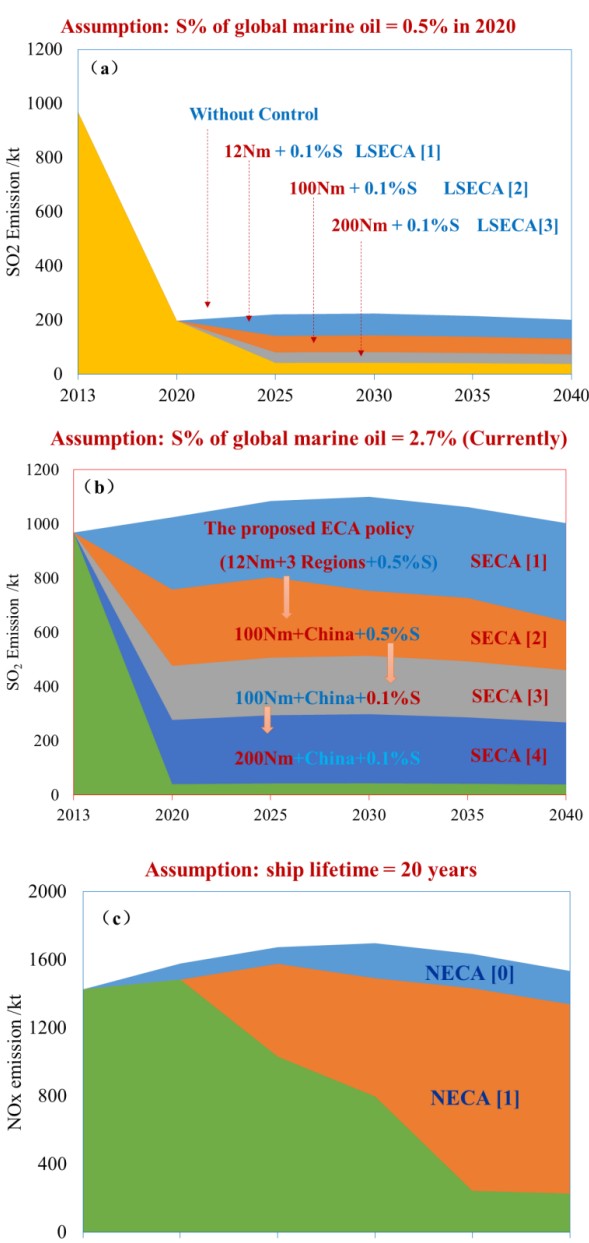

**Fig. 10 SO₂ and NOx emissions from the shipping sector under SO₂-ECA (a, b) and NOx-ECA (c) with different control policies from 2013 to 2040**





**Table 1 Time-in-mode for different ship types within 12 Nm and 200 Nm of the coast**

**(unit: hours/voyage)**

| Ship types | | Cruising | | | | | | Maneuvering | Hotelling |
|---|---|---|---|---|---|---|---|---|---|
| | | Bohai | Shang dong | YRD | Taiwan Strait | PRD | Beibu Gulf | China | China |
| **200Nm** | | | | | | | | | |
| OGVs | Tanker | 64.1 | 61.4 | 73.4 | 40.5 | 73.3 | 46.8 | 4.3 | 25.3 |
| | Cargo | 53.4 | 51.2 | 61.1 | 33.7 | 61.1 | 39.0 | 3.4 | 15.8 |
| | Container | 41.8 | 40.1 | 47.8 | 26.4 | 47.8 | 30.6 | 3.7 | 22.2 |
| | Others | 53.1 | 50.9 | 60.8 | 33.5 | 60.7 | 38.8 | 1.1 | 17.2 |
| CVs | Tanker | 52.2 | 36.5 | 47.9 | 37.5 | 53.6 | 42.4 | 2.3 | 23.5 |
| | Cargo | 45.3 | 31.7 | 41.5 | 32.5 | 46.5 | 36.8 | 3.2 | 16.8 |
| | Container | 37.7 | 26.4 | 34.6 | 27.1 | 38.7 | 30.6 | 3.9 | 19.1 |
| | Others | 45.1 | 31.5 | 41.3 | 32.4 | 46.3 | 36.6 | 2.7 | 17.7 |
| **12Nm** | | | | | | | | | |
| OGVs | Tanker | 20.9 | 11.8 | 20.4 | 14.4 | 19.6 | 9.2 | 4.3 | 25.3 |
| | Cargo | 17.4 | 9.8 | 17.0 | 12.0 | 16.3 | 7.7 | 3.4 | 15.8 |
| | Container | 13.6 | 7.7 | 13.3 | 9.4 | 12.8 | 6.0 | 3.7 | 22.2 |
| | Others | 17.3 | 9.8 | 16.9 | 11.9 | 16.2 | 7.6 | 1.1 | 17.2 |
| CVs | Tanker | 16.8 | 9.0 | 16.5 | 11.2 | 15.7 | 6.7 | 2.3 | 23.5 |
| | Cargo | 14.6 | 7.8 | 14.3 | 9.7 | 13.6 | 5.8 | 3.2 | 16.8 |
| | Container | 12.2 | 6.5 | 11.9 | 8.1 | 11.4 | 4.9 | 3.9 | 19.1 |
| | Others | 14.5 | 7.8 | 14.2 | 9.7 | 13.6 | 5.8 | 2.7 | 17.7 |
| RVs | All ships | 2.3 | | | | | | 7.5 | 23.3 |

**Table 2 Transportation distance for CVs and OGVs in different regions (unit: Nm)**

| Regions[a] | OGVs within 12Nm | OGVs within 200Nm | CVs within 12Nm | CVs within 200Nm |
|---|---|---|---|---|
| Bohai Rim | 313 | 961 | 219 | 679 |
| Shandong Province | 177 | 921 | 117 | 475 |
| Yangtze River Delta | 306 | 1100 | 214 | 622 |
| Western Taiwan Strait | 216 | 607 | 146 | 487 |
| Pearl River Delta | 293 | 1099 | 204 | 697 |
| Beibu Gulf | 138 | 703 | 88 | 551 |

5     [a]The average transport distance of ships arriving in/departing the corresponding port region.





**Table 3 Summary of ship calls number by marine department in 2013**

| Marine Department[a] | OGV | CV | RV | Total |
|---|---|---|---|---|
| Liaoning | 9541 | 31618 | 56520 | 97679 |
| Hebei | 4904 | 22935 | 165 | 28004 |
| Tianjin | 8780 | 13871 | 0 | 22651 |
| Shandong | 15097 | 20040 | 2231 | 37368 |
| Jiangsu | 12886 | 17802 | 906648 | 937336 |
| Shanghai | 18592 | 23835 | 623587 | 666014 |
| Zhejiang | 17915 | 81759 | 398653 | 498327 |
| Fujian | 10317 | 38911 | 85777 | 135005 |
| Guangdong | 14281 | 65681 | 1270203 | 1350165 |
| Shenzhen[b] | 13704 | 7166 | 122186 | 143056 |
| HongKong[c] | 11672 | 15404 | 77374 | 104450 |
| Hainan[d] | 2475 | 9466 | 10869 | 22810 |
| Sum | 140164 | 348488 | 3554213 | 4042865 |

[a]Ordered from north to south China.
[b]the MD of Shenzhen was independent of the Guangdong MD;
[c]refers to the HKMD official website.
[d]the domain corresponding to statistics compiled by the Hainan MD covered the ports in Hainan and
Guangxi provinces.

**Table 4 Sulfur transfer rates of different engine types used in this study**

| Engine types | USEPA, 2006 | IMO, 2014 | Fan et al., 2016 | This study[a] |
|---|---|---|---|---|
| SSD | 82.9% | 82.9% | 81.8% | 82.5% ± 0.5% |
| MSD | 90.2% | 91.4% | 89.8% | 90.5% ± 0.7% |
| HSD | 96.6% | 96.5% | 89.8% | 94.3% ± 3.2% |

[a]Calculated based on the average values reported in previous studies.



**Table 5 Ship emissions reduction analysis scenarios of SO₂-DECA and NOx-DECA**

| Scenario | | Description | Code |
|---|---|---|---|
| | | No DECAs control | SECA[0] |
| SO₂-DECA | International oil quality standards would be achieved (S%=0.5%) by 2020 | Shipping in DECA should use low sulfur content fuel( S%=0.1% )within 12Nm | LSECA[1] |
| | | Shipping in DECA should use low sulfur content fuel( S%=0.1% )within 100Nm | LSECA[2] |
| | | Shipping in DECA should use low sulfur content fuel( S%=0.1% )within 200Nm | LSECA[3] |
| | Improve the international oil quality (S%=2.7%) | Implement the DECA policy proposed in 2016; shipping in Bohai, YRD and PRD region within 12Nm should use low Sulphur content fuel (S%=0.5%) | SECA[1] |
| | | Shipping in all of China within 100Nm should use low Sulfur content fuel (S%=0.5%) | SECA[2] |
| | | Shipping in all of China within 100Nm should use low Sulfur content fuel (S%=0.1%) | SECA[3] |
| | | Shipping in all of China within 200Nm should use low Sulfur content fuel (S%=0.1%) | SECA[4] |
| NOx-DECA | | No controls, namely NOx controls under 'Tier 0', 'Tier I', 'Tier II' | NECA[0] |
| | | Create NOx-DECAs under 'Tier III' | NECA[1] |

**Table 6 Comparison of MHO consumption in China in 2013 (unit: kt)**

| Sources | Domains | OGVs | CVs | RVs | Total |
|---|---|---|---|---|---|
| This work:MDO-M1/M2[a] | 12 Nm[c] | 310/317 | 703/732 | 1156/1390 | 2170/2439 |
| This work:MHO-M1/M2[a] | 12 Nm[c] | 3690/4157 | 1814/2102 | 0/0 | 5504/6260 |
| This work:MDO-M1/M2[a] | 140 Nm[c] | 410/336 | 1007/849 | 1156/1390 | 2573/2574 |
| This work:MHO-M1/M2[a] | 140 Nm[c] | 8368/8801 | 5380/5258 | 0/0 | 13748/14059 |
| This work:MDO-M1/M2[a] | 200 Nm[c] | 456/492 | 1007/849 | 1156/1390 | 2620/2730 |
| This work:MHO-M1/M2[a] | 200 Nm[c] | 10946/11777 | 5380/5258 | 0/0 | 16326/17035 |
| Liu et al., 2016[b] | China sea | 22455 | | -- | 22455 |
| Liu et al., 2016[b] | East Asian sea | 35087 | | -- | 35087 |
| China Energy Statistic Yearbook | Sales of MHO in transport sector | -- | -- | -- | 15888 |

5  [a]M1/M2 represent cargo-based and port-based approach, respectively; MDO and MHO represent marine diesel oil and marine Heavy oil.

[b]Calculated using CO₂ emissions and an emission factor of 3591.12 g CO₂/ kg of Fuel (see IMO Report, 2009).

[c]The distance to Chinese shore.



**Table 7 Port activity and fuel consumption in the top ten ports in China**

| Port | Cargo throughput /10 kt | Container throughput /10 kTEU | # of ship calls /VAN | MHO/ kt | | MDO/ kt | |
|---|---|---|---|---|---|---|---|
| | | | | FC within 12 Nm | FC in port | FC within 12 Nm | FC in port |
| Shanghai | 68273 | 3362 | 666018 | 1058 | 296 | 410 | 213 |
| Ningbo-Zhoushan | 80978 | 1735 | 401164 | 1027 | 222 | 170 | 100 |
| Hong Kong | 27606 | 2235 | 104450 | 590 | 143 | 110 | 36 |
| Guangzhou | 45517 | 1531 | 479229 | 520 | 140 | 370 | 138 |
| Shenzhen | 23398 | 2328 | 143052 | 531 | 140 | 120 | 50 |
| Tianjing | 50063 | 1301 | 22651 | 345 | 54 | 0 | 0 |
| Suzhou | 45435 | 531 | 105795 | 271 | 59 | 260 | 26 |
| Dalian | 40746 | 1001 | 40465 | 225 | 39 | 90 | 10 |
| Xiamen | 19088 | 801 | 56670 | 184 | 44 | 76 | 13 |
| Qingdao | 45003 | 1552 | 14481 | 146 | 38 | 60 | 0 |
| Sum | 446107 | 16376 | 2033975 | 4897 | 1176 | 1665 | 586 |
| All ports | 1176705 | 19021 | 4042865 | 6260 | 1826 | 2439 | 1024 |
| Sum/All | 38% | 86% | 50% | 78% | 64% | 68% | 57% |




**Table 8 Summary of ship emissions in China (within 200 Nm of the coast) in 2013**

| Categories | | SO$_2$ | NOx | CO | PM$_{10}$ | PM$_{2.5}$ | VOCs |
|---|---|---|---|---|---|---|---|
| Ship types | Cargo ship | 38% | 41% | 42% | 39% | 41% | 40% |
| | Container | 39% | 37% | 37% | 39% | 37% | 39% |
| | Tanker | 17% | 15% | 15% | 16% | 17% | 14% |
| | Others | 6% | 7% | 7% | 5% | 6% | 7% |
| Engine types | Main engine | 79% | 79% | 80% | 79% | 81% | 82% |
| | Auxiliary engine | 20% | 20% | 20% | 20% | 18% | 17% |
| | Auxiliary boiler | 1% | 1% | 1% | 1% | 1% | 1% |
| Ship trade types | OGVs | 67% | 63% | 63% | 68% | 67% | 57% |
| | CVs | 31% | 31% | 32% | 30% | 31% | 30% |
| | RVs | 2% | 6% | 5% | 2% | 3% | 12% |
| Activity modes | Cruising | 79% | 79% | 69% | 76% | 76% | 65% |
| | Maneuvering | 5% | 6% | 13% | 8% | 8% | 19% |
| | Hotelling | 16% | 15% | 18% | 16% | 17% | 16% |
| Total/ kt | | 1010 | 1443 | 118 | 107 | 87 | 67 |

**Table 9 Uncertainties in emission estimates**

| | Species | Emission estimate (kt) | Mean (kt) | 95%CI (kt) | Uncertainty | Previous studies | |
|---|---|---|---|---|---|---|---|
| | | | | | | Li et al. (2016) | Liu et al. (2016) |
| Port-based | SO$_2$ | 1010 | 972 | (911, 1058) | (-6.3%, 8.8%) | (-21.2%, 28.6%) | (-3.8%, 3.8%) |
| | NO$_X$ | 1443 | 1433 | (1348, 1556) | (-5.9%, 8.6%) | (-22.1%, 30.6%) | (-3.6%, 3.6%) |
| | CO | 118 | 122 | (109, 143) | (-10.4%, 18.1%) | (-22.6%, 30.3%) | (-4.6%, 4.6%) |
| | PM$_{10}$ | 107 | 113 | (103, 128) | (-8.4%, 13.5%) | (-22.7%, 30.7%) | (-3.8%, 3.8%)[a] |
| | PM$_{2.5}$ | 87 | 98 | (89, 111) | (-8.4%, 13.5%) | (-22.8%, 31.5%) | -- |
| | VOCs | 67 | 73 | (66, 82) | (-8.6%, 13.8%) | (-24.5%, 33.3%) | (-4.0%, 4.0%)[b] |

[a, b] were the results of PM and NMVOC, respectively.





**Table 10 Emission intensities in three DECAs and along typical navigation lines**

| | | DECA | | | Topical Transport Line[a] | | | | Total within 200 Nm |
|---|---|---|---|---|---|---|---|---|---|
| | | Bohai-DECA | YRD-DECA | PRD-DECA | ① Bohai &YRD | ② PRD &YRD | ③ Korea &YRD | ④ Janp & PRD | |
| Sea area ($10^4$ km$^2$) | | 7.70 | 5.35 | 2.29 | 2.62 | 5.90 | 2.55 | 2.99 | 156.56 |
| Emissions (kt) | SO$_2$ | 73 | 190 | 104 | 22 | 180 | 26 | 12 | 1010 |
| | NOx | 107 | 283 | 156 | 33 | 270 | 39 | 18 | 1443 |
| | PM$_{2.5}$ | 8 | 22 | 12 | 2 | 18 | 3 | 1 | 107 |
| Intensity (t/km$^2$) | SO$_2$ | 0.94 | 3.55 | 3.07 | 0.83 | 3.05 | 1.03 | 0.41 | 0.65 |
| | NOx | 1.39 | 5.29 | 4.60 | 1.25 | 4.57 | 1.54 | 0.61 | 0.92 |
| | PM$_{2.5}$ | 0.11 | 0.42 | 0.36 | 0.08 | 0.31 | 0.10 | 0.04 | 0.07 |

[a] Geographic location of transport lines shown in Fig. SI-5.