# Peer review of "Manuscript under review for journal Atmos. Chem. Phys."

_Atmospheric Chemistry and Physics, 2017_

## Referee Comment (RC1) · Anonymous Referee #2 · 24 Nov 2017

This manuscript touches an important field and shows interesting results. Both the backward estimation and the forecasting for future years are important for policy making. Establishing an integrated ship emission estimation and validation approach will enhance the understanding of activity and emissions as well. There are a few comments that need to be addressed to improve the paper. The most important, to make the calculation solid and transparent to readers. It's very hard to evaluate the accuracy of results based on limited information provided in paper. Maybe it's mainly due to the unclear description. The following questions need to be answered and described in detail for this revision. 1. The cargo-based approach is very unclear. How do you get emissions other than 2013? This method is the key for the whole paper. The authors

use only ten lines to give a very brief description. Without detailed data, it's hard to prove the results are convinced. 1) I suggest to list all the data in tables. 2) What is the transport volume? Is it based on port statistic? 3) How many ports with transport volume do you have? How do you generate regional transport volume based on port statistics? 4) Do you considered those ship only pass the region without a destination in that region? If those ships were overlooked, are the results still reliable? 5) How do you define the transport distance? With AIS information only, you cannot get the origin and destination of each trip. Fig. SI-1 didn't explain how you get the distance. 6) Do you mean that all the cargo share the same transport distance? Is it true? 7) Section 2.3.2. No data was provided at all! How can I evaluate your calculation results without any input data? You can decide to provide data in tables or delete all the related results. 8) The data source should be clearly provided in linkage or with DOI. Such general description, such as "China yearbook", means nothing to most of the audients who can not read Chinese! 2. How's the quality of the AIS database? It seems the authors make calculation based on very limited AIS data. 1) Page 6, Line 10-13, I was confused by the two methods you mentioned. Monthly variation is not from AIS? You have only one day per month for AIS? 2) If so, is there large weekly or monthly variation of shipping activity in China? 3) Page 7, line 26, only 700 AIS-based trajectories from 2013? That means, you have two trajectories for each day. If so, how can you estimate emissions from other ships? 3. The ship information database is far from enough. Only 5000 ships from LRS and 7600 RVs from local MDs were collected. How many ships were observed in your AIS or port calls database? You may lose most of the ships by doing this. You mentioned another study reported 18000 ships, which actually cannot support your study. If you want to catch 20000 ships travelling in to your study domain, you need to prepared a database much larger than this number. So the missing ships in your database is at least more than half. 4. What is the boundary of China sea area? Do 200nm regions all belong to China? Without a boundary, the emissions can't be considered as the China's emissions. 5. What is the definition of China's emission control area? Through the manuscript, it seems the 200 nm is defined as the emission

control zone. It seems very strange to me. The zone is extended to other countries, like Viet Nam, Korea and etc. 6. Page 14, line 8-11, why the fuel consumption were in a good agreement with those in cargo and container turnover? In the past 30 years in US, the fuel consumption increased much slower than the cargo turnover. Because the ship fleet gets larger and more fuel efficient. Actually, authors calculated fuel consumption based on the cargo turnover. It's your assumption the fuel should be in the same trend with cargo, not a conclusion. So, this conclusion is not correct. 7. All the abbreviations should be listed with full name when first appeared, and using only abbreviations for latter. Such as MHO, line 18 Page 8, line 12 page 9. There are a lot of these errors... 8. Usually we don't say marine heavy oil, but HFO (heavy fuel oil). 9. The format of references need to be checked carefully. Some references missed information, eg. journal name. For example: Li C., Yuan Z.B., Ou J.M., Fan X.L., Ye S.Q., Xiao T., Shi Y.Q., Huang Z.J., Ng S.K.W., Zhong Z.M., and Zheng J.Y.. 2016. An AIS-based high-resolution ship emission inventory and its uncertainty in Pearl River Delta region, China. 573:1-10. http://dx.doi.org/10.1016/j.scitotenv.2016.07.219 For journal names, both the abbreviation and full name were used, eg. Atmosphere Chemical Physic and Atmos. Chem. Phys. 10. Figure 2 and 3 should be put into supplementary information.

---

## Referee Comment (RC2) · Anonymous Referee #3 · 11 Jan 2018

The study investigates the ship emissions along the cost of mainland China. The paper is well written and organized, with the approach clearly described and the results fully discussed. The ship emission inventory developed in this study has an high grid resolution and a long time period coverage. Such emission estimates along with dataset are interesting and should be a welcome addition to both global and regional emission inventories existed. The manuscript can be accepted for publication in ACP subject to minor revisions. Below are my questions/suggestions in detail.

Both global and regional (Asian) ship emission inventories have been developed before this study. The advantage(s) of the ship emission inventory developed in this study over

[Figure]

previous ones are not sufficiently highlighted in the manuscript. In the Conclusions, the authors may give some suggestions for the modelers and other users who want to make a choice among different ship emission inventories.

BC and OC are important components for air quality, visibility and climate simulations, and they are included in nearly all emission inventories for modeling purpose. However, BC and OC are not considered (or reported) in this work. Is it easy to add these two components?

Technical issues: P3, L13: The literature (He at al., 2015) cannot be found in the References list.

P4, L4: Better to provide specific names of the three classification schemes.

P26, Fig.5a: Should the line colors of the cost match those in the pie? Green color (for YRD in the pie) cannot be found in the lines for the cost.

P28, Fig.7: The current color bar is not clear to see. Are the ship emissions associated with Taiwan ports taken into account? How about the emissions over the South China Sea?

The term 'HC' is used in the text while the term 'VOC' is given in Tables 8 and 9. There are so many abbreviations used in the manuscript. The authors might consider giving a list of abbreviations as Appendix?
* * *

---

## Author Comment (AC1) · 21 Feb 2018

Feb 13, 2018

Prof. Junyu Zheng
Jinan University
Guangzhou 511442, China
Tel: +86-20-37336618, Fax: +86-20-37336618
E-mail: zhengjunyu_work@hotmail.com

Attn: Atmospheric Chemistry & Physics

Dear Prof. Ma:

Enclosed is a revised manuscript submitted for consideration of publication in Atmospheric Chemistry & Physics. The manuscript title is *Decadal evolution of ship emissions in China from 2004 to 2013 by using an integrated AIS-based approach and projection to 2040*.

We thank both reviewers for their valuable comments. This manuscript has been improved significantly by addressing these comments, and a detailed point-by-point response to all the comments is provided. All revisions made in the manuscript that correspond to comments are listed with line numbers to facilitate future examinations. Specifically, we have detailed the cargo-based approach in determining cargo volume and transport distance, complemented the major raw data and explained the representative of AIS data, added to calculate the results of black carbon (BC) and organic carbon (OC), and highlight the advantages of our approaches and results.

Should you have any questions to this work, please do not hesitate to contact us.

Yours sincerely,

Junyu Zheng
Professor
Institute for Environmental and Climate Research

---

## Author Comment (AC2) · 21 Feb 2018

1. The cargo-based approach is very unclear. How do you get emissions other than 2013? This method is the key for the whole paper. The authors use only ten lines to give a very brief description. Without detailed data, it's hard to prove the results are convinced. Response: In equation (2), cargo-based approach is to estimate emissions by transport volume, transport distance, fuel consumption, and emission factor. In the revised manuscript, we explain briefly the methods in determining transport volume and transport distance in lines 4-21 of page 6, and provide more detailed explanation in section 3 and 4 of the support information (SI), respectively. Determination of

fuel consumption rate and emission factor are introduced in section 2.3.2 and 2.3.3, respectively.

1.1 I suggest to list all the data in tables. Response: We list detailed transport volume data for all 100 ports in Table SI-5, and transport distance for ocean-going vessels (OGVs) and coast vessel (CVs) in Table 2, more data for transport distance calculation in Table SI-6 and Figure SI-1(b).

1.2 What is the transport volume? Is it based on port statistic? How many ports with transport volume do you have? How do you generate regional transport volume based on port statistics? Response: Transport volume is the real weight of transport cargo for a period time. Yes, it is based on port statistics and was extracted from the statistic yearbook of 100 Chinese ports in 6 port clusters. Due to the lack of transport volume in difference ship types, the stock of waterway cargo types in different provinces was separated into OGVs, CVs and RVs using the province-specific throughput of coastal ports and river ports, and it was then adjusted by the contributions of foreign trade in the main ports. Additional, the regional transport volume statistics include liquid cargo, dry bulk, general cargo, and container, corresponding to tanker, bulk ship, general cargo ship, container ship, respectively. Regional transport volume is determined by classifying ports into port clusters (regions). There are six port clusters in this study, including Bohai, Shandong, YRD, Western Taiwan Strait, PRD and Beibu Gulf. We list detailed transport volume data for all 100 ports in Table SI-5. The above information shown in lines 5-10 of page 6 and section 3 of the SI.

1.3 Do you considered those ship only pass the region without a destination in that region? If those ships were overlooked, are the results still reliable? Response: We did roughly estimate the contribution from passing ships, and concluded that their contribution is relatively low but with potentially high uncertainties. Therefore, we decide to exclude it into this study to avoid negative impact on the results. The research domain is 200Nm to the coast of Mainland China. The main routes in this domain include all routes from/to Chinese ports and the passing routes, mainly from Busan, Korea

to Southeast Asia (Busan route) and from Taiwan to destinations other than Mainland China ports (Taiwan route). In order to study the fraction of Busan route and Taiwan route in our research domain, we extracted a real-time AIS map, and highlighted the passing ship routes by red lines. There were 368 shipping route from/to Korea in 2013, including 85 Southeast Asia routes and 26 Europe routes. As the throughput of Busan port accounted for 75.4% of total throughput (17686kt) in Korea, we estimated that Busan route roughly accounted for 7100kt throughput. With around 800Nm passing distance in our research domain, we estimated the fuel consumption from Busan route was around 70kt HFO. The total throughput in Taiwan was 14 million TEU in 2013, including 2.5 million TEU between Taiwan and Mainland China. Therefore, Taiwan route contributed around 11.5 million TEU. If we assume 1TEU=15t and the average travel distance was 500Nm, the fuel consumption from Taiwan route was around 1070kt HFO. Therefore, the total consumption of Busan route and Taiwan route was around 1140kt HFO, only 7% of total fuel consumption in our research domain. Therefore, we believe excluding the passing route would not significantly impact our analysis results. We briefly mention the exclusion of passing route in lines 14-15 of page 4 of the manuscript, and provide detailed explanation in section 2 of SI.

Fig. SI-1 Major shipping routes extracted from a real-time AIS digital map (passing routes are highlighted in red)

1.4 How do you define the transport distance? With AIS information only, you cannot get the origin and destination of each trip. Fig. SI-1 didn't explain how you get the distance. Response: Transport distance is the weight-based length along common routes of OGVs and CVs in the research domains of 12Nm and 200Nm, respectively. Specifically, transport distances of OGVs were calculated as the average of main international routes from main ports in a particular port cluster, as shown in Fig. SI-2(a), and then multiply by the fraction of regular routes to Korea, Japan, South China Sea and Pacific, respectively (see Table SI-6); transport distances of CVs were derived from transport distances between port clusters measured by AIS data and digital map. Some illustrations are given in Fig. SI-2(c) for readers to understand the AIS-based digital map. We collected information more than 1000 regular routes, including their departure and arrival ports. We classified departure and arrival ports into port clusters, and then used AIS data and digital map to calculate transport distances between port clusters (Fig. SI-2(b)). It should be emphasized that the departure and arrival port information for the regular route information is not collected by AIS data. AIS data is only used to calculate inter-port cluster transport distance. The above information shown in lines 10-19 of page 6 and section 4 of the SI.

1.5 do you mean that all the cargo share the same transport distance? Is it true? Response: This is not true. Table 2 lists transport distances for OGVs and CVs in different regions within 12Nm and 200Nm calculated from more than 1000 shipping routes. Detailed calculation procedure of transport distances for OGVs and CVs are provided in lines 10-19 of page 6. We didn't calculate transport distance of RVs as we directly use fuel consumption of 5.2 tce/10Kt provided by Statistics Communique of China on the Traffic and Transportation Industry Development, as shown in section 7 of the SI.

1.6 Section 2.3.2. No data was provided at all! How can I evaluate your calculation results without any input data? You can decide to provide data in tables or delete all the related results. Response: We add detailed data about transport volume in Table SI-5 and transport distance in Table SI-6(b) and Figure SI-1(b), respectively. We also provide detailed explanation on the calculation process in sections 3 and 4 of the SI.

1.7 The data source should be clearly provided in linkage or with DOI. Such general description, such as "China yearbook", means nothing to most of the audients who can not read Chinese! Response: The yearbook is only published in Chinese. The linkages are provided in the reference list. Unfortunately, some raw data, such as vessel calling number and cargo volume, was provided by local marine departments without any linkage or reference material.

2. How's the quality of the AIS database? It seems the authors make calculation based on very limited AIS data. Response: AIS database used in this study is limited in number but with high representativeness, because of 1) this study is not aimed to calculate emissions from each AIS data point, therefore does not need all AIS data; 2) the objective of using AIS data in this study is to accurately identify ship activity characteristics and main parameters, e.g. transport distance, time-in-mode, loading factor, therefore it is a must that AIS data matches well with source categories in the emission inventory; 3) strict selection criteria were adopted. First, ship categories and weight tonnage of OGVs, CVs and RVs were analyzed (Fig. SI-4). The number of ship route in different weight tonnages for different ship categories were then calculated. Prefer to frequent active ships within this study domain, 700 AIS trajectories with high representativeness were then selected; 4) In comparison with Liu et al. (2016) which used East Asia as target domain, the area of research domain in this study was around 71% and covered 69% of ship information, including all regular routes. This meant that the density of ship information in this study was comparable with Liu et al. (2016). Therefore, we believe the AIS database used in this study, although with relatively limited number, had indeed high quality and representativeness. The above information is added in lines 11-13 of page 8 and section 5 of the SI.

2.1 Page 6, Line 10-13, I was confused by the two methods you mentioned. Monthly variation is not from AIS? You have only one day per month for AIS? If so, is there large weekly or monthly variation of shipping activity in China? Response: Monthly variation of fuel consumption is derived from cargo throughput instead of AIS mainly for the following two reasons. 1) Within a year, the structure of cargo types and transport routes in a particular port generally don't have significant changes. Therefore, ship activity has significant correlation with cargo throughput. Monthly variation of cargo throughput can be used as a substitute for monthly variation of fuel consumption; 2) Monthly variations of fuel consumption in different years tend to be different as they are largely affected by the fluctuation of international trade. In this study, we used monthly variation of cargo throughput from 2000 to 2013 to account for such annual variation.

In addition, we examined the density distribution of AIS data and found that the weekly and daily variations were not obvious, as shown in Fig. SI-3. Therefore, it is reasonable to use data from one day to account for monthly variation. The above information is added in lines 26-31 of page 6 and section 5 of the SI.

2.2 Page 7, line 26, only 700 AIS-based trajectories from 2013? That means, you have two trajectories for each day. If so, how can you estimate emissions from other ships? Response: No. We have the entire one-year data for all 700 trajectories. We have revised the expression as "approximately 700 AIS-based navigation trajectories with entire one-year data from 2013 were collected" to avoid misunderstanding. The above information is revised in lines 8 of page 8.

3. The ship information database is far from enough. Only 5000 ships from LRS and 7600 RVs from local MDs were collected. How many ships were observed in your AIS or port calls database? You may lose most of the ships by doing this. You mentioned another study reported 18000 ships, which actually cannot support your study. If you want to catch 20000 ships travelling in to your study domain, you need to prepare a database much larger than this number. So the missing ships in your database is at least more than half. Response: Our AIS database included 700 trajectories and port calls database included 9.54 million port calls, including 6.58 million for RVs and 2.96 million for OGVs and CVs. In comparison with a previous study (Liu et al., 2016) which used East Asia as target domain, the area of research domain in this study was around 71% and covered 69% of ship information, including all regular routes. This meant that the density of ship information in this study was comparable with Liu et al. (2016). Therefore, we believe the AIS and ship databases used in this study, although with relatively limited number, had indeed high quality and representativeness. Another objective of this study is establish a methodology in using limited AIS data to develop ship emission inventory. Such a methodology can be used in other parts of the world, as most of the time it is unable to collect a complete set of AIS information. The above information is added in lines 11-16 of page 8, Table 3, and section 5 of the SI.

4. What is the boundary of China Sea area? Do 200nm regions all belong to China? Without a boundary, the emissions can't be considered as the China's emissions. What is the definition of China's emission control area? Through the manuscript, it seems the 200 nm is defined as the emission control zone. It seems very strange to me. The zone is extended to other countries, like Viet Nam, Korea and etc. Response: Current domestic emission control areas (DECAs) in China only covers 12Nm in three main port clusters. The main target of this study is to investigate the effect of DECA delineation in assisting emission control, therefore used 200Nm as one of the scenarios, solely for research purpose. We selected 200Nm for two reasons. 1) 200Nm is the border for exclusive economic zones (EEZ), and 2) 200 Nm offshore has been officially approved by International Maritime Organization (IMO) and been used by North America ECA. We want to emphasize that our 200Nm scenario does not include territorial sea of other countries, even if it is within 200Nm offshore of China, as illustrated in Fig.1. In addition, the setting of our research domain does not involve any political consideration. The above information is added in lines 30-31 of page 3 and lines 3-5 of page 4.

5. Page 14, line 8-11, why the fuel consumption were in a good agreement with those in cargo and container turnover? In the past 30 years in US, the fuel consumption increased much slower than the cargo turnover. Because the ship fleet gets larger and more fuel efficient. Actually, authors calculated fuel consumption based on the cargo turnover. It's your assumption the fuel should be in the same trend with cargo, not a conclusion. So, this conclusion is not correct. Response: We made a mistake here. Fig. 6 shows that the fuel consumption doubled during 2004-2013, and container and cargo transport volume almost tripled as shown in Fig. 7. The relevant sentence has been revised in lines 19-24 of page 14.

6. All the abbreviations should be listed with full name when first appeared, and using only abbreviations for latter. Such as MHO, line 18 Page 8, line 12 page 9. There are a lot of these errors. . . Response: Revised accordingly.

7. Usually we don't say marine heavy oil, but HFO (heavy fuel oil). Response: Revised

accordingly.

8. The format of references need to be checked carefully. Some references missed information, e.g. journal name. For example: Li C., Yuan Z.B., Ou J.M., Fan X.L., Ye S.Q., Xiao T., Shi Y.Q., Huang Z.J., Ng S.K.W., Zhong Z.M., and Zheng J.Y.. 2016. An AIS-based high-resolution ship emission inventory and its uncertainty in Pearl River Delta region, China. 573:1-10. http://dx.doi.org/10.1016/j.scitotenv.2016.07.219 For journal names, both the abbreviation and full name were used, e.g. Atmosphere Chemical Physic and Atmos. Chem. Phys. Response: Revised accordingly.

9. Figure 2 and 3 should be put into supplementary information. Response: Revised accordingly.   Reference: Liu H., Fu M.L., Jin X.X., Shang Y. Shindell D., Faluvegi G., Shindell C., He K.B., 2016. Health and climate impacts of ocean-going vessels in East Asia. Nature climate change. doi: 10.1038/NCLIMATE3083.

---

## Author Comment (AC3) · 21 Feb 2018

1. The advantage(s) of the ship emission inventory developed in this study over previous ones are not sufficiently highlighted in the manuscript. In the Conclusions, the authors may give some suggestions for the modelers and other users who want to make a choice among different ship emission inventories. Response: The advantages of ship emission inventory are threefold. 1) We used two different methods (cargo-based and port-based) to estimate and mutually validate emissions; 2) We calculated the ten-year trend of ship emissions from 2004-2013, and made projections in different scenarios with implementation of DECA; 3) We established a methodology in using

limited AIS data to develop ship emission inventory. Such a methodology can be used in other parts of the world, as most of the time it is unable to collect a complete set of AIS information. The above advantages are discussed in the Implication section of the manuscript. The above information is added in lines 31-32 of page 18, and lines 1-4 of page 19.

2. BC and OC are important components for air quality, visibility and climate simulations, and they are included in nearly all emission inventories for modeling purpose. However, BC and OC are not considered (or reported) in this work. Is it easy to add these two components? Response: We added BC and OC analysis in the revision.

3. P3, L13: The literature (He at al., 2015) cannot be found in the References list. Response: We instead provide three peer-reviewed publications for the Multi-resolution Emission Inventory for China (MEIC) (Li et al., 2014; Zheng et al., 2014; Liu et al., 2015).

4. P4, L4: Better to provide specific names of the three classification schemes. Response: Specific names are provided in the revision.

5. P26, Fig.5a: Should the line colors of the coast match those in the pie? Green color (for YRD in the pie) cannot be found in the lines for the coast. Response: Revised accordingly.

6. P28, Fig.7: The current color bar is not clear to see. Are the ship emissions associated with Taiwan ports taken into account? How about the emissions over the South China Sea? Response: We have changed the color bar to become more visually clear. Taiwan ports were not taken into account due to the absence complete data sources. Our research domain only covered 200Nm offshore, therefore didn't account for the entire South China Sea.

7. The term 'HC' is used in the text while the term 'VOC' is given in Tables 8 and 9. There are so many abbreviations used in the manuscript. The authors might consider

giving a list of abbreviations as Appendix? Response: All VOCs have been changed to HCs. We also provide a list of abbreviations as Appendix, as suggested.

  Reference: Li, M., Zhang, Q., Streets, D. G., He, K. B., Cheng, Y. F., Emmons, L. K., Huo, H., Kang, S. C., Lu, Z., Shao, M., Su, H., Yu, X., and Zhang, Y.. 2014. Mapping Asian anthropogenic emissions of non-methane volatile organic compounds to multiple chemical mechanisms, Atmospheric Chemistry & Physics, 14, 5617–5638, doi:10.5194/acp-14-5617-2014. Liu, F., Zhang, Q., Tong, D., Zheng, B., Li, M., Huo, H., and He, K. B.. 2015. High-resolution inventory of technologies, activities, and emissions of coal-fired power plants in China from 1990 to 2010, Atmospheric Chemistry & Physics, 15, 13299–13317, doi:10.5194/acp-15-13299-2015. Zheng, B., Huo, H., Zhang, Q., Yao, Z. L., Wang, X. T., Yang, X. F., Liu, H., and He, K. B.. 2014. High-resolution mapping of vehicle emissions in China in 2008, Atmospheric Chemistry & Physics, 14, 9787–9805, doi:10.5194/acp-14-9787-2014.

---

## Author Comment (AC4) · 21 Feb 2018

**Decadal evolution of ship emissions in China from 2004 to 2013 by using an integrated AIS-based approach and projection to 2040**

Cheng Li1, Jens Borken-Kleefeld2, Junyu Zheng1, Zibing Yuan3, Jiamin Ou4, Yue Li5, Yanlong Wang3, Yuanqian Xu3

[revised manuscript text omitted]

---

## Author Comment (AC5) · 21 Feb 2018

Support information change with red marked.

Please also note the supplement to this comment:
https://www.atmos-chem-phys-discuss.net/acp-2017-743/acp-2017-743-AC5-supplement.pdf
* * *